# HIF-1α Promotes Luteinization via NDRG1 Induction in the Human Ovary

**DOI:** 10.3390/biomedicines13020328

**Published:** 2025-01-31

**Authors:** Akemi Nishigaki, Mitsuaki Ishida, Hiroaki Tsubokura, Yoji Hisamatsu, Yoshinobu Hirose, Hidetaka Okada

**Affiliations:** 1Department of Obstetrics and Gynecology, Kansai Medical University, Hirakata 573-1191, Osaka, Japan; nishiake@hirakata.kmu.ac.jp (A.N.); h.tsubokura26@gmail.com (H.T.); hisamayj@hirakata.kmu.ac.jp (Y.H.); hokada@hirakata.kmu.ac.jp (H.O.); 2Department of Pathology, Osaka Medical and Pharmaceutical University, Takatsuki 569-8686, Osaka, Japan; yoshinobu.hirose@ompu.ac.jp

**Keywords:** hypoxia, hypoxia-inducible factor-1α, N-myc downstream-regulated gene 1, ovary, steroidogenesis

## Abstract

**Background/Objectives:** Hypoxia-inducible factor-1α (HIF-1α) is a transcription factor that plays a crucial role in various physiological and pathological processes of the ovary. However, the timing of HIF-1α expression and its specific biological function in the follicular development of the human ovary remain unclear. Therefore, in this study, we aimed to examine whether HIF-1α and its downstream gene, *N-myc downstream-regulated gene 1* (*NDRG1*), exhibit stage-specific expression during the follicular development process in the human ovary. **Methods**: We used ovarian tissues from eight women with regular menstrual cycles who were not undergoing hormonal treatment. We investigated HIF-1α and NDRG1 expression and localization using immunohistochemistry. Further, we transfected human ovarian granulosa (KGN) cells with *HIF-1α* small interfering RNA (siRNA) to investigate the influence of *HIF-1α* on *NDRG1* expression and progesterone synthesis. **Results**: The immunohistochemical analysis of human ovarian tissues revealed that HIF-1α was localized in the cytoplasm of granulosa cells (GCs) at both the primary and secondary follicular stages. Conversely, in tertiary and later developmental stages, HIF-1α was observed exclusively in the nucleus of GCs. Furthermore, while NDRG1 was not detected in primary follicles, it was present in all GCs beyond the tertiary stage. Notably, transfection of KGN cells with *HIF-1α* siRNA significantly decreased NDRG1 expression, at both the mRNA and protein levels, and in progesterone synthesis. **Conclusion**: Our results indicate that HIF-1α and NDRG1 are integral to follicular development and the early luteinization of pre-ovulatory follicles.

## 1. Introduction

During the process of follicular growth, the follicle undergoes various structural changes, including alterations in the vascular system, cell differentiation, and proliferation, along with the development of a fluid-filled antrum [1]. Collectively, these transformations establish a hypoxic environment within the follicle, where the oocyte develops under low-oxygen conditions [2,3]. Hypoxia-inducible factor-1α (HIF-1α) is the main mediator of cellular response to hypoxia, which acts as an oxygen-regulated transcriptional activator [4]. HIF-1α is an essential regulator in the development of various physiological systems and plays a crucial role in maintaining tissue homeostasis, because it influences cell survival and adaptation, immune responses, cytokine secretion, anaerobic metabolism, and angiogenesis [5,6,7,8,9]. The HIF-1α protein stabilizes under hypoxic conditions and is translocated from the cytoplasm to the nucleus [10], where it forms a dimer with HIF-1β [11]. This heterodimer then binds to hypoxia response elements in the promoters of target genes, activating the transcription of HIF-regulated genes involved in numerous physiological processes [4,11,12].

HIF-1α has been identified in the granulosa cells (GCs) of various species such as humans, monkeys, pigs, rats, mice, and cows [13,14,15,16,17,18]. Recent research has highlighted HIF-1α as a significant regulator of gene expression within ovarian compartments, contributing to physiological follicle development. For example, Zhang et al. reported that HIF-1α expression is decreased in atretic follicles compared to mature healthy antral follicles of pigs [19]. HIF-1α is also essential for ovulation as inhibiting its function leads to anovulation in mice [18]. Additionally, HIF-1α is a key regulator of vascular endothelial growth factor A expression, which is important for angiogenesis and luteal formation [3,20]. Collectively, substantial evidence is available, which suggests that HIF-1α regulates various genes in GCs and is involved in the development and differentiation of ovarian follicles in mammals. However, the expression and functions of HIF-1α during human ovarian development remain incompletely understood.

N-myc downstream-regulated gene 1 (NDRG1) plays important roles in various biological processes such as angiogenesis, cell proliferation, differentiation, apoptosis, as well as tumor progression and metastasis in response to various stress signals such as hypoxia, oxidative stress, and hormones [21,22,23,24,25,26,27]. Additionally, its expression is notably increased by several transcription factors linked to hypoxia, particularly HIF-1α, under low-oxygen conditions, in various types of tissues [28,29,30]. NDRG1 is mainly located in the cytoplasm and is widely distributed across different tissues [21,31]. In the human ovary, NDRG1 expression has been detected in oocytes, GCs, theca-interstitial cells, and the corpus luteum [28]. Notably, our previous study has shown that reducing *NDRG1* levels leads to the decreased mRNA expression of *steroidogenic acute regulator protein (StAR)* and *CYP11A1* (*p450scc),* as well as a reduction in P_4_ production, which suggests that NDRG1 is involved in progesterone synthesis in human GCs [28].

Given the diverse functions of HIF-1α, we hypothesized that it is involved in the development of the ovarian follicles in humans and that its localization varies throughout the different phases of the follicular cycle. Therefore, in this study, we aimed to clarify the localization of the HIF-1α protein at various stages of human ovarian follicle development. Additionally, we examined how HIF-1α affects the expression of its downstream gene *NDRG1* and the production of P4 in GCs. Exploring the functions of HIF-1α in the human ovary could enhance our understanding of follicle development and luteinization.

## 2. Materials and Methods

### 2.1. Patient Selection

The ovarian tissues utilized in this research were collected from eight women (ages 32–39) with regular menstrual cycles, who were not undergoing hormonal treatment for uterine cancer, at Kansai Medical University Hospital between January and December 2021. Formalin-fixed and paraffin-embedded whole ovarian tissues were used for immunohistochemical analysis. All human samples were obtained with written informed consent from each participant, in accordance with the Declaration of Helsinki. This study received approval from the Ethics Review Board of Kansai Medical University, Osaka, Japan (project approval number 2008105).

### 2.2. Immunohistochemistry for HIF-1α and NDRG1

We evaluated the developmental stages of follicles (i.e., primary, secondary, and tertiary follicles, and corpus luteum) in the ovarian tissues because they contain follicles at various developmental stages. HIF-1α and NDRG1 protein expression was examined at each follicular developmental stage. Immunohistochemical analysis was performed using an auto-stainer (Discovery ULTRA System; Roche Diagnostics, Basel, Switzerland; Leica BOND-III; Leica Biosystems GmbH, Nussloch, Germany) according to the manufacturer’s instructions. A primary rabbit polyclonal antibody was used to detect HIF-1α (1:200; cat. no. ab51608; Abcam, Cambridge, UK) and NDRG1 (1:75; Cell Signaling Technology, Danvers, MA, USA). Antigen retrieval was performed by autoclaving tissue sections at 100 °C for 16 min in a Tris-based buffer (pH 8.5; Cell Conditioning Solution CC1; Ventana Medical Systems, Oro Valley, AZ, USA). Thereafter, automated protocol steps for immunostaining were followed, using 3,3′-diaminobenzidine (DAB) as a colorimetric agent.

### 2.3. Cell Culture

The KGN cell line (RCB1154) was acquired from the RIKEN Cell Bank of Japan (Tsukuba, Japan) and was cultured in Dulbecco’s modified Eagle’s medium (DMEM)/F-12 supplemented with 10% fetal calf serum (FCS; HyClone, Logan, UT, USA), 100 U/mL penicillin, and 100 μg/mL streptomycin (Life Technologies, Thermo Fisher Scientific, Waltham, MA, USA) in an atmosphere of 5% CO_2_ at 37 °C. Luteinized GCs were prepared by incubating KGN cells with 10 µM forskolin (Selleckchem, Huston, TX, USA), which strongly increases cAMP and stimulates steroid synthesis by mimicking the action of gonadotropins. All experiments were performed at least in triplicate with cell preparations each time.

### 2.4. HIF-1α Silencing with Small Interfering RNA (siRNA)

The silencing of *HIF-1α* expression was achieved using siRNA preparations directed against distinct regions of the gene sequence (Human Stealth Select RNAi; Life Technologies, Carlsbad, CA, USA). Stealth siRNA targeting *HIF-1α* (Catalog numbers: HSS104775 and HSS104774) as well as a non-silencing RNA negative control (Catalog number: 12935300 Human Stealth Select RNAi™) were used. The cells were allowed to grow until they reached approximately 30% confluency in DMEM/F12 medium supplemented with 10% FCS, as determined by optimizing the experimental conditions. Subsequently, they were transfected with each siRNA (10 nmol/L) using Lipofectamine™ RNAiMAX transfection reagent (Invitrogen, Carlsbad, CA, USA) according to the manufacturer’s guidelines for 72 h. KGN cells were transfected with or without 10 μM forskolin in DMEM/F12 supplemented with 10% dextran-coated stripped FCS for progesterone measurements.

### 2.5. RNA Extraction and Quantitative Reverse-Transcription PCR (RT-PCR)

Total RNA was extracted from cultured KGN cells using an RNeasy Minikit (Qiagen GmbH, Hilden, Germany). cDNA was synthesized from 1 µg of RNA using ReverTra Ace qPCR RT Master Mix (Toyobo, Osaka, Japan) according to the manufacturer’s instructions. Quantitative RT-PCR was conducted using Rotor-Gene Q HRM (Qiagen GmbH, Hilden, Germany) along with a quantitative PCR mix kit (THUNDERBIRD SYBR qPCR Mix; Toyobo, Osaka, Japan), following a previously established protocol [32]. The Ct method was used to calculate the relative mRNA expression. Ct values were standardized to *Elongation factor-1α* (*EF-1α*) as an internal control. The primer sequences used in quantitative RT-PCR are provided in Table 1.

### 2.6. Western Blot Analysis

Cultured cells were homogenized using lysis buffer containing RIPA buffer (Cell Signaling Technology, Danvers, MA, USA) and phenylmethanesulfonyl fluoride (Cell Signaling Technology, Danvers, MA, USA) to assess protein levels. The samples were then centrifuged at 14,000× *g* to collect the cell pellets. Protein concentrations were quantified using a Bio-Rad Protein Assay (Bio-Rad Laboratories, Hercules, CA, USA). Equal amounts of lysates (20 μg/lane) were separated via electrophoresis on an Any kD™ Mini-PROTEAN^®^ TGX™ Precast Protein Gel and were electro-transferred onto nitrocellulose membranes using a Trans-Blot Turbo device (Bio-Rad Laboratories, Hercules, CA, USA). The membranes were blocked with 10% skim milk powder dissolved in Tris-buffered saline for 1 h to minimize nonspecific binding. The blots were then incubated overnight at 4 °C with the primary antibodies, including rabbit polyclonal NDRG1 antibody (1:1000; cat. no. #5196: Cell Signaling Technology, Danvers, MA, USA), rabbit monoclonal StAR antibody (1:1000; cat. no. #8449: Cell Signaling Technology, Danvers, MA, USA), rabbit monoclonal CYP11A1 (p450scc) antibody (1:1000; cat. no. #14217: Cell Signaling Technology, Danvers, MA, USA), or mouse monoclonal β-actin antibody (1:5000; cat. no. A5316: Sigma-Aldrich, St. Louis, MO, USA). Subsequently, the blots were probed with anti-rabbit immunoglobulin IgG (1:5000; GE Healthcare Life Sciences, Chicago, IL, USA) or anti-mouse IgG (1:10,000; GE Healthcare Life Sciences, Chicago, IL, USA) peroxidase-conjugated secondary antibodies. Enhanced chemiluminescence plus Western blotting detection agents (GE Healthcare Life Sciences, Chicago, Il, USA) were used to visualize immune complexes.

### 2.7. Measurement of Hormone Secretion

The concentration of P_4_ in cell culture medium was measured using commercially available ELISA Kits specific for progesterone (Cayman Chemical Co., Ann Arbor, MI, USA) according to the manufacturer’s instructions. Data are presented as the quantity of steroids secreted (pg/mL).

### 2.8. Statistical Analysis

Data are presented as mean ± standard error of the mean (SEM). Multiple comparisons were analyzed using one-way analysis of variance (ANOVA), followed by Dunnett’s or Tukey’s test for multiple comparisons. GraphPad Prism 8 was used to conduct statistical analyses (GraphPad Software, Inc., La Jolla, CA, USA). Statistical significance was set at *p* < 0.05.

## 3. Results

### 3.1. Localization of HIF-1α Protein Expression in the Human Follicles

We performed immunohistochemical analyses to determine the localization of HIF-1α protein expression within the human ovary (Figure 1). Table 2 shows the positive rates of HIF-1α protein at various ovarian follicular development stages. Immunohistochemical analysis of human ovarian tissue sections revealed positive HIF-1α staining in the GCs of ovarian follicles at various developmental stages. HIF-1α protein was detected in the cytoplasm of more than 70% of GCs in primary (Figure 1A) and secondary follicles (Figure 1B), whereas it was barely detectable in the nucleus of GCs. Conversely, in tertiary follicles, HIF-1α was predominantly localized in the nucleus of GCs, with no significant cytoplasmic staining observed, and its expression was reduced in the corpus luteum (Figure 1C,D).

### 3.2. Localization of NDRG1 Protein Expression in the Human Follicles

We performed immunohistochemical analyses to investigate the localization of NDRG1 protein expression in the human ovary (Figure 2). Table 3 shows the positive rates of the NDRG1 protein at various follicular developmental stages. NDRG1 expression was not observed in the nucleus at all stages of follicular development and was restricted to the cytoplasm of GCs. NDRG1 protein was also detected in the cytoplasm of theca-interstitial cells. NDRG1 protein expression was not detected in the cytoplasm of GCs in primary follicles (Figure 2A). In contrast, NDRG1 expression was observed in some secondary follicles (Figure 2B) and was present in all GCs at stages beyond tertiary follicles (Figure 2C,D). Thus, NDRG1 protein expression increased with follicular development.

### 3.3. Regulation of NDRG1 Expression by HIF-1α

We subsequently investigated the impact of HIF-1α on NDRG1 expression by transfecting KGN cells with *HIF-1α* siRNA to explore the interaction between these two proteins. We confirmed the efficiency of *HIF-1α* knockdown using quantitative RT-PCR, which revealed a significant reduction in *HIF-1α* mRNA levels 24 h after siRNA transfection (Figure 3A). After 72 h of *HIF-1α* siRNA transfection, KGN cells exhibited a significant decrease in the expression of NDRG1 both at the mRNA and protein levels (Figure 3B,C).

### 3.4. Effect of HIF-1α Silencing on Hormone Production

We further investigated the effect of siRNA-mediated knockdown of *HIF-1α* expression on hormone production to directly analyze the critical role of HIF-1α in hormone production. The addition of FSK significantly increased the expression of StAR, CYP11A1 (p450scc), and progesterone. After 72 h of *HIF-1α* siRNA transfection, KGN cells exhibited a notable decrease in StAR and CYP11A1 (p450scc) mRNA and protein levels (Figure 4A–C). Moreover, a notable reduction was observed in the release of P_4_ into the culture medium of these cells (Figure 4D).

## 4. Discussion

In this study, we demonstrated that HIF-1α is primarily expressed in the GCs of the human ovary, with its expression levels and patterns varying significantly across different developmental stages in humans. NDRG1 was expressed at a slightly later stage than HIF-1α and was detected in all GCs at stages beyond tertiary follicles. Furthermore, HIF-1α plays a role in the regulation of luteinization-related genes and P_4_ secretion, highlighting the importance of HIF-1α signaling in ovarian folliculogenesis and luteinization.

During the growth and maturation of ovarian follicles, the microenvironment within the follicles is characterized by hypoxic condition [1,3]. Specifically, oxygen levels in the follicular fluid of the human ovary decrease as folliculogenesis advances [2]. Notably, the ovarian follicle is adapted to function effectively under low-oxygen conditions. Cellular adaptations to low oxygen levels are primarily mediated by HIF-1α [4]. HIF-1α regulates the expression of various target genes and is vital for numerous biological processes, including ovarian functions [7,8,9,20,28,33,34]. Although the roles of HIF-1α in folliculogenesis have recently been investigated in various species [13,14,15,16,17,18,19], its expression in the human ovary has not been well characterized. Therefore, we investigated the expression and localization of HIF-1α protein in human ovaries at different phases of follicular development. We observed changes in HIF-1α immunolocalization across the ovarian cycle, suggesting its involvement in the follicular development. Specifically, we found that HIF-1α protein was localized in the cytoplasm of primary and secondary follicles. HIF-1α was weakly localized in the nucleus of primary and secondary follicles, whereas strong nuclear localization was observed in tertiary follicles. Consistent with our results, another study using mammalian cell lines has indicated that HIF-1α expression is mainly cytoplasmic under normoxic conditions, whereas it localizes to the nucleus under hypoxic conditions [10]. Consequently, the low-oxygen environment during follicular development is suggested to cause HIF-1α to translocate into the nucleus. Moreover, in our study, we observed that HIF-1α expression in the corpus luteum was decreased. Notably, HIF-1α expression decreases over time after ovulation in various species [14,15,16,35], and the changes in HIF-1α expression during follicular development suggest its critical role in follicular growth and maturation. In addition, we observed a lack of HIF-1α staining in some tertiary follicles, which corroborates the finding by Zhang et al., who demonstrated that HIF-1α expression was downregulated in atretic follicles compared to mature healthy antral follicles [19]. Overall, the absence of HIF-1α expression may indicate follicular terminal differentiation.

We previously showed that NDRG1 was expressed in human GCs and the corpus luteum [28]. However, to our knowledge, the expression patterns of NDRG1 and the relationship between NDRG1 and HIF-1α proteins expressions throughout ovarian stages have not been extensively studied in any species, including humans. Accordingly, in the present study, we utilized immunohistochemistry to analyze relative NDRG1 protein expression and noted differences between NDRG1 and HIF-1α expression at various stages of follicular development. Specifically, NDRG1 was not expressed in primary follicles but was expressed in some secondary follicles and highly expressed in tertiary follicles and the corpus luteum. NDRG1 was also expressed at a later stage than HIF-1α, which corresponds to the stage at which HIF translocates to the nucleus. Furthermore, we elucidated its role in follicular development in the ovary by knocking down HIF-1α expression. This knockdown resulted in the downregulation of NDRG1 expression at both the mRNA and protein levels. Our previous study demonstrated that echinomycin inhibits hypoxia-induced NDRG1 expression by specifically blocking the DNA-binding capability of HIF-1α through direct interaction with the hypoxia response element of its target gene [28]. This explains our observations of decreased NDRG1 expression in both normoxic and hypoxic environments following *HIF-1α* knockdown. Collectively, these observations suggest that HIF-1α is a key initiator of NDRG1 expression and that its signaling pathway plays a significant role in regulating NDRG1 expression in GCs.

In this study, the transfection of KGN cells with *HIF-1α* siRNA resulted in a significant decrease in the mRNA and protein levels of StAR and CYP11A1 (p450scc). StAR is responsible for transporting cholesterol from the outer to the inner mitochondrial membrane, while CYP11A1 (p450scc) catalyzes the reaction that converts cholesterol to pregnenolone, which represents an early and limiting step in steroid hormone synthesis [36]. Moreover, StAR is crucial for the synthesis of P_4_ and the development and maintenance of the corpus luteum in humans and other primates [37]. The expression of StAR in GCs indicates the early functional maturation of ovarian antral follicles [38]. Thus, our results are consistent with previous studies [17,20,34,39] that identified HIF-1α as a contributor to StAR expression. Overall, these results suggest that HIF-1α facilitates steroidogenesis and luteinization, which is the terminal differentiation process of GCs in the ovary. After ovulation, GCs undergo luteinization to form the corpus luteum and secrete P_4_, which is essential for establishing pregnancy [40,41]. In fact, our results highlight the role of HIF-1α in promoting luteinization, since silencing HIF-1α resulted in decreased P4 secretion. Therefore, these results provide strong evidence that HIF-1α plays a crucial role in promoting P_4_ production during luteinization in the human ovary. Nonetheless, additional in vivo studies are necessary for clarifying the specific functions of HIF-1α in folliculogenesis and luteinization. In our previous study, we explored the effect of NDRG1 in hormone production and demonstrated that NDRG1 is involved in P_4_ production but does not affect estrogen production. In addition, we found that NDRG1 knockdown results in a reduction in *StAR* and *CYP11A1* (*p450scc*) mRNA expression [28]. In the present study, we showed that *HIF-1α* knockdown results in a decrease in both *NDRG1* expression and P4 production. These findings indicate that HIF-1α plays a role in inducing NDRG1 expression and that NDRG1, in turn, regulates the expression of StAR and p450scc, facilitating P4 production. Consequently, our results imply that HIF is crucial for the regulation of NDRG1 expression in the ovary, with both HIF-1α and NDRG1 contributing to the luteinization process (Figure 5).

In summary, this study is the first to reveal significant changes in HIF-1α expression throughout the various developmental stages of the human ovary. This suggests that HIF-1α signaling is a key mechanism in regulating folliculogenesis in humans, although further exploration is required to fully understand the underlying mechanisms. However, the *HIF-1α* knockdown experiments conducted in this study were performed only under normoxic conditions. Therefore, future studies should investigate the effects of hypoxia on the HIF signaling pathway in the ovary. Such investigations may have implications for future infertility treatments and could yield new insights into ovarian physiology, including the processes of ovarian folliculogenesis.

## Figures and Tables

**Figure 1 biomedicines-13-00328-f001:**
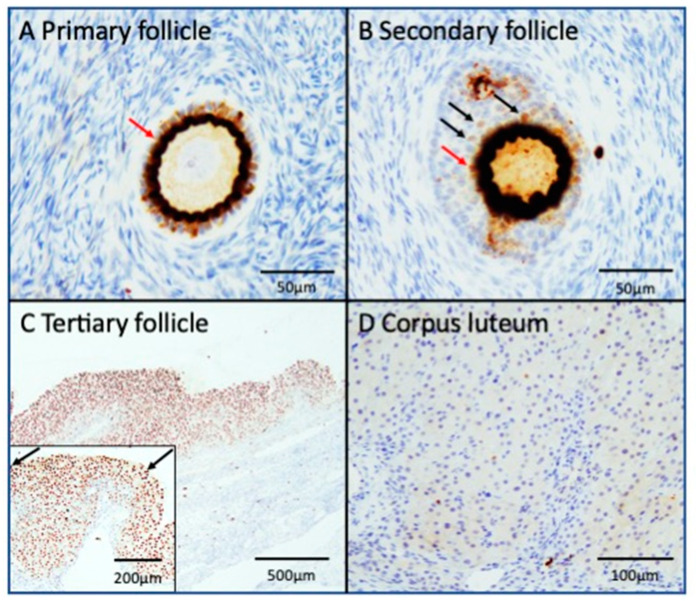
Immunolocalization of HIF-1α expression in the human follicle. The immunolocalization of HIF-1α in human follicles is depicted, with representative data from eight specimens. (**A**) Primary follicle. (**B**) Secondary follicle. (**C**) Tertiary follicle. (**D**) Corpus luteum. The cytoplasm of granulosa cells in primary and secondary follicles exhibited positive staining with anti-HIF-1α antibody (indicated by red arrows). In the tertiary and later stages of development, HIF-1α expression was confined to the nucleus (indicated by black arrows in the high power field of the tertiary follicle ((**C**) inset)).HIF-1α, hypoxia-inducible factor-1α.

**Figure 2 biomedicines-13-00328-f002:**
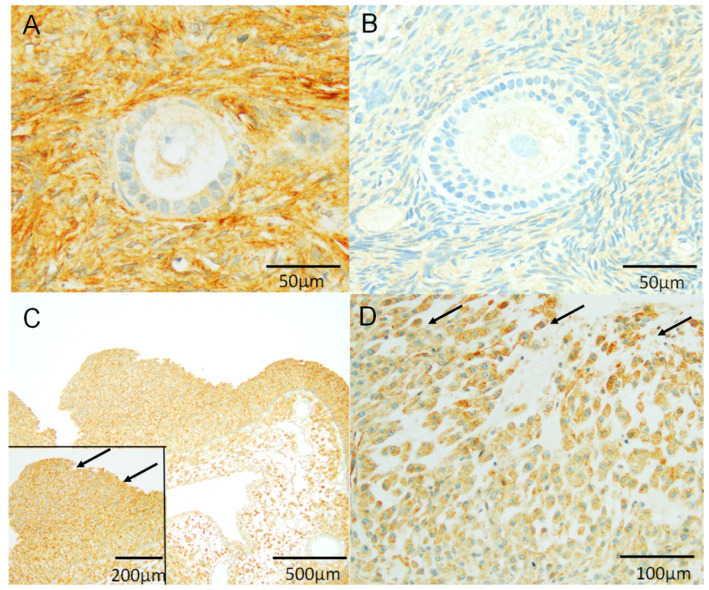
Immunolocalization of NDRG1 expression in the human follicle. The immunolocalization of NDRG1 in human follicles is depicted, with representative data from eight specimens. (**A**) Primary follicle. (**B**) Secondary follicle. (**C**) Tertiary follicle. (**D**) Corpus luteum. The cytoplasm of the GCs was positively stained with anti-NDRG1 antibody in the tertiary follicle and corpus luteum (arrows) (the high power field of the tertiary follicle ((**C**) inset). NDRG1, N-myc downstream-regulated gene 1.

**Figure 3 biomedicines-13-00328-f003:**
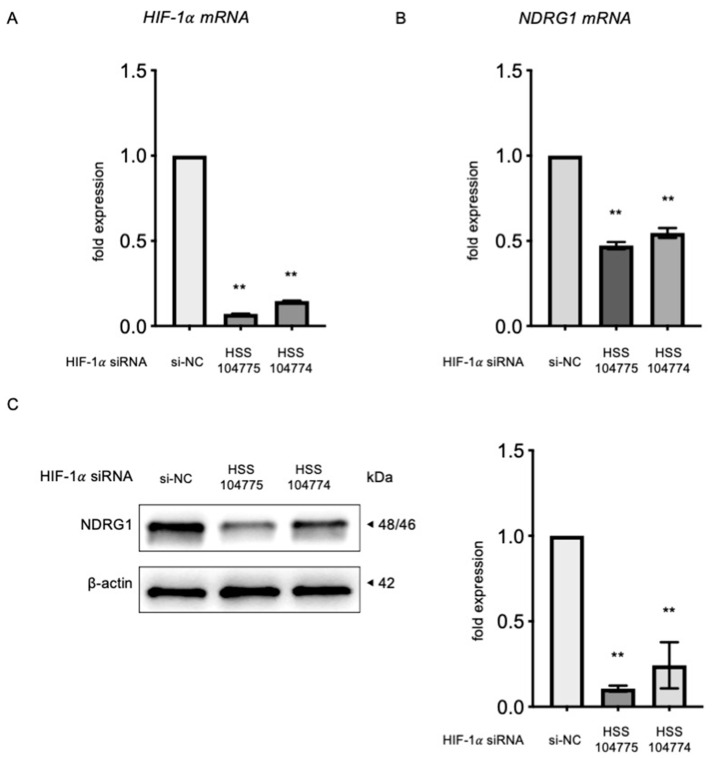
Effect of *HIF-1α* silencing on NDRG1. (**A**) Effect of *HIF-1α* silencing on *HIF-1α* mRNA. KGN cells were transfected with either siRNA targeting HIF-1α or a control siRNA for 24 h, and the efficiency of *HIF-1α* knockdown was assessed using quantitative reverse-transcription polymerase chain reaction (RT-PCR) (*n* = 6). Effect of HIF-1α silencing on *NDRG1* mRNA (**B**) and protein (**C**). KGN cells were transfected with HIF-1α siRNA or control siRNA for 72 h. The effect of *HIF-1α* knockdown on *NDRG*1 mRNA expression was evaluated using quantitative RT-PCR (*n* = 6). The protein levels of NDRG1 were analyzed through Western blotting, with β-actin serving as a loading control (*n* = 3). At least three experiments were conducted for each. The results of densitometric analysis were normalized to the expression of β-actin to ensure accuracy in quantification. The data were presented as mean ± SEM. ** *p* < 0.01 versus the si-NC group. HSS104774 and HSS104775 are siRNA IDs. NDRG1, N-myc downstream-regulated gene 1; HIF-1α, hypoxia-inducible factor-1α; RT-PCR, reverse-transcription polymerase chain reaction.

**Figure 4 biomedicines-13-00328-f004:**
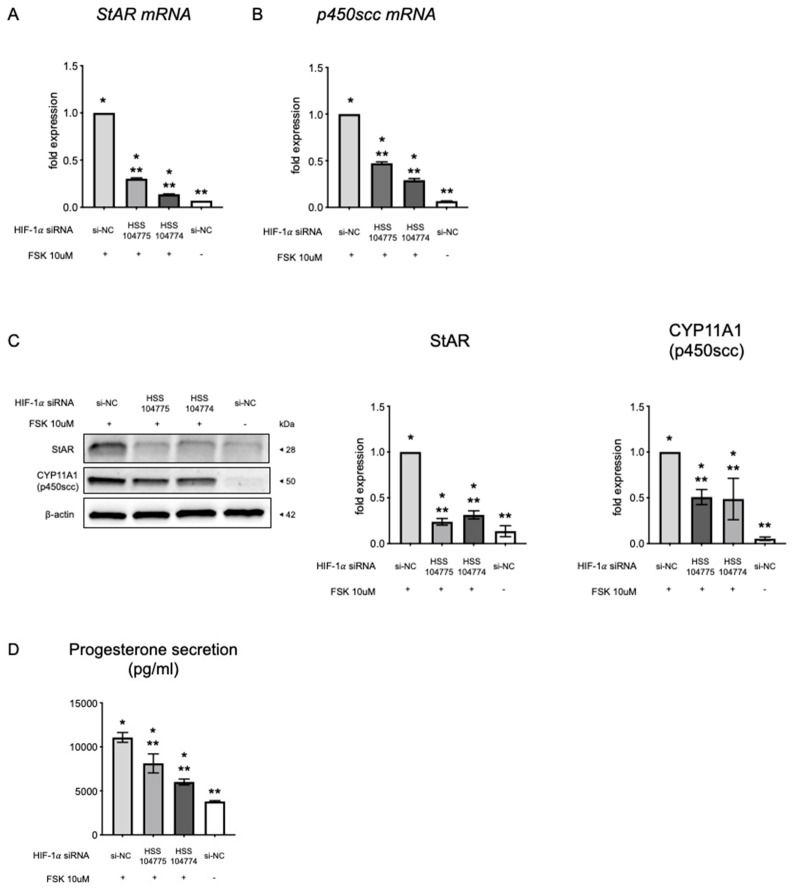
Effect of *HIF-1α* silencing on hormone production. Effect of HIF-1α silencing on *StAR* mRNA (**A**), *p450scc* mRNA (**B**), and StAR and CYP11A1 (p450scc) protein (**C**). KGN cells were transfected with either siRNA targeting *HIF-1α* or a control siRNA for 72 h. The effect of *HIF-1α* knockdown was evaluated using quantitative RT-PCR (*n* = 6). NDRG1 protein levels were quantified using Western blotting, with β-actin serving as a control (*n* = 3). At least three experiments were conducted for each. Densitometric analysis results were normalized to the expression of β-actin. (**D**) Effect of *HIF-1α* silencing on progesterone secretion. Progesterone concentration in the cell culture supernatant was analyzed using the enzyme immunoassay protocol after 72 h. Data are presented as the amount of steroids secreted, with results expressed as the means ± SEM. * *p* < 0.01 versus the si-NC group. ** *p* < 0.01 versus the si-NC + FSK group. HSS104774 and HSS104775 are siRNA IDs. FSK, forskolin; HIF-1α, hypoxia-inducible factor-1α; RT-PCR, reverse-transcription polymerase chain reaction; StAR, steroidogenic acute regulator protein.

**Figure 5 biomedicines-13-00328-f005:**
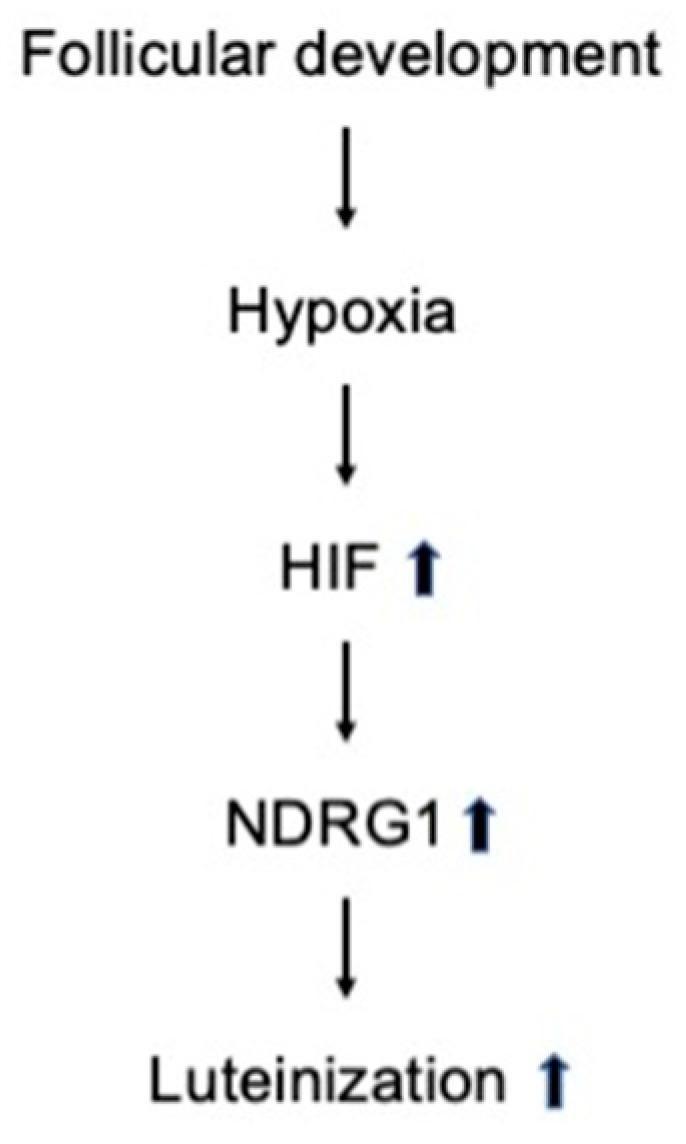
Schematic representation of the potential mechanisms involved in luteinization. The hypoxic environment during follicular development induces the stabilization of HIF-1α. HIF mediates NDRG1 and promotes luteinization.

**Table 1 biomedicines-13-00328-t001:** List of primer sequences for quantitative RT-PCR.

Gene	Forward Primer (5′→3′)	Reverse Primer (5′→3′)
HIF-1α	TTCACCTGAGCCTAATAGTCC	CAAGTCTAAATCTGTGTCCTG
NDRG1	AGGCAGGTGACAGCAGGGAC	CGTGGCAGACGGCAAAGT
StAR	AAACTTACGTGGCTACTCAGCATC	GACCTGGTTGATGATGCTCTTG
p450scc	CAGGAGGGGTGGACACGAC	AGGTTGCGTGCCATCTCATAC
EF	TCTGGTTGGAATGGTGACAACATGC	AGAGCTTCACTCAAAGCTTCATGG

**Table 2 biomedicines-13-00328-t002:** Immunohistochemical staining for HIF-1α in ovarian follicles.

	Number of Samples			
	Total	HIF-1α +	HIF-1α -	Positive Rate (%)
**Cytoplasmic**				
primary follicles	69	54	15	78.3
secondary follicles	11	8	3	72.7
tertiary follicle	18	0	18	0.0
corpus luteum	13	0	13	0.0
**Nuclear**				
primary follicles	69	1	68	1.4
secondary follicles	11	1	10	9.1
tertiary follicle	18	13	5	72.2
corpus luteum	13	2	11	15.4

**Table 3 biomedicines-13-00328-t003:** Immunohistochemical staining for NDRG1 in ovarian follicles.

	Number of Samples			
	Total	NDRG1 +	NDRG1 -	Positive Rate (%)
primary follicles	68	0	68	0.0
secondary follicles	10	5	5	50.0
tertiary follicle	6	6	0	100.0
corpus luteum	12	12	0	100.0

## Data Availability

The original contributions presented in this study are included in this article. Further inquiries can be directed to the corresponding author(s).

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
