# Peer review of "HIF-1α Promotes Luteinization via NDRG1 Induction in the Human Ovary"

_biomedicines, 2025, doi:10.3390/biomedicines13020328_

Round 1
Reviewer 1 Report
Comments and Suggestions for Authors
In this study, the authors detected the expressions of NDRG1 and HIF-1α in the human ovary and explore the effect of HIF-1α on the expression of NDRG1 and progesterone synthesis in granulosa cells. However, the experimental design was r simple and lacked a rigorous approach, making it difficult to draw possible conclusions.
1. The title of this manuscript does not adequately reflect the content of the study.
2.According to the staining results, NDRG1 is mainly expressed in the stromal cells of the ovary, rather than in the granulosa cells.
3.The effect of NDRG1 interference needs to be validated at the protein level.
4. The expression of P450scc and StAR were also examined at the protein level.
5. Does NDRG1 mediate the regulatory effect of HIF on progesterone? Relevant experimental designs are lacking.
Author Response
Point-by-point responses to reviewer’s comments:
Thank you for reviewing our manuscript. We appreciate your constructive comments, which were very helpful in revising our manuscript. We have revised the manuscript following the concerns highlighted by you.
Reviewer #1
In this study, the authors detected the expressions of NDRG1 and HIF-1α in the human ovary and explore the effect of HIF-1α on the expression of NDRG1 and progesterone synthesis in granulosa cells. However, the experimental design was r simple and lacked a rigorous approach, making it difficult to draw possible conclusions.
- The title of this manuscript does not adequately reflect the content of the study.
Response:
Thank you for your pertinent comment. As per your suggestion, we have revised the title to better reflect the content of the present study.
“HIF-1α promotes luteinization via NDRG1 induction in the human ovary”
2.According to the staining results, NDRG1 is mainly expressed in the stromal cells of the ovary, rather than in the granulosa cells.
Response:
As you mentioned, stromal cells of the ovary were positive for NDRG1. We added the comment regarding NDRG1 expression in the stromal cells of the ovary.
Page 6 line 194
“NDRG1 protein was also detected in the cytoplasm of theca interstitial cells.”
3.The effect of NDRG1 interference needs to be validated at the protein level.
Response:
Thank you for your pertinent comment. As per your suggestion, we have included the results of NDRG1 protein expression level via knockdown of HIF-1α in Figure 3C. Additionally, we have revised the text as follows:
Page 7 lines 213–215
“After 72 h of HIF-1α siRNA transfection, KGN cells exhibited a significant decrease in the expression of NDRG1 both at the mRNA and protein levels (Figure 3B, C).”
- The expression of P450scc and StAR were also examined at the protein level.
Response:
As per your suggestion, we performed additional experiments to assess the protein levels of StAR and CYP11A1 (p450scc). Accordingly, we have included the following text, in addition to Figure 4C.
Page 7–8 lines 231–234
“After 72 h of HIF-1α siRNA transfection, KGN cells exhibited a notable decrease in StAR and CYP11A1 (p450scc) mRNA and protein levels (Figures 4A, B, C).”
- Does NDRG1 mediate the regulatory effect of HIF on progesterone? Relevant experimental designs are lacking.
Response:
Thank you for your pertinent comment. In the present study, our results demonstrate that transfection of KGN cells with HIF-1α siRNA significantly decreased NDRG1, StAR, and CYP11A1 (p450scc) expression, at both the mRNA and protein levels, and in progesterone synthesis. In addition, we previously showed that the knockdown of NDRG1 mRNA results in a reduction in StAR and CYP11A1 (p450scc) mRNA expression. Moreover, the immunohistochemical staining results of the present study revealed that HIF-1α activation was the preceding event to NDRG1 expression in human follicular development. Therefore, HIF-1α is crucial for the regulation of NDRG1 expression in the ovary, with both HIF-1α and NDRG1 contributing to the luteinization process, although the other regulating mechanism regarding HIF-1α and NDRG1 activation cannot be completely excluded. We have added the following text explaining this in Discussion.
Page 10 lines 323–326
“In our previous study, we explored the effect of NDRG1 in hormone production and demonstrated that NDRG1 is involved in P4 production but does not affect estrogen production. In addition, we found that NDRG1 knockdown results in a reduction in StAR and CYP11A1 (p450scc) mRNA expression.”
Reviewer 2 Report
Comments and Suggestions for Authors
The research addresses a gap in our understanding of the role of hypoxia-inducible factor-1α (HIF-1α) and its target gene NDRG1 in the follicular development process in the human ovary. This study is methodologically robust and provides valuable insights into the stage-specific expression of HIF-1α and NDRG1 during different follicular stages. The use of immunohistochemistry and RNA interference provides compelling evidence that these factors may play a critical role in the progression of follicular development, particularly at the transition to luteinisation.
Only a minor revision is required:
It would enhance the discussion to include a more detailed explanation of how forskolin affects steroidogenesis and why it was chosen as a control in the experiment.
Author Response
Reviewer #2
Thank you for reviewing our manuscript. We appreciate your constructive comments, which were very helpful in revising our manuscript. We have revised the manuscript following the concerns highlighted by you.
The research addresses a gap in our understanding of the role of hypoxia-inducible factor-1α (HIF-1α) and its target gene NDRG1 in the follicular development process in the human ovary. This study is methodologically robust and provides valuable insights into the stage-specific expression of HIF-1α and NDRG1 during different follicular stages. The use of immunohistochemistry and RNA interference provides compelling evidence that these factors may play a critical role in the progression of follicular development, particularly at the transition to luteinisation.
Only a minor revision is required:
It would enhance the discussion to include a more detailed explanation of how forskolin affects steroidogenesis and why it was chosen as a control in the experiment.
Response:
Thank you for your pertinent comment. As per your suggestion, we added an explanation regarding the use of forskolin.
Page 3 lines 107–110
“Luteinized GCs were prepared by incubating KGN cells with 10 µM forskolin (Selleckchem, Huston, TX, USA), which strongly increases cAMP and stimulates steroid synthesis by mimicking the action of gonadotropins. ”
Reviewer 3 Report
Comments and Suggestions for Authors
The present study aimed to analyse the functions of HIF-1α and NDRG1 in various stages of human ovary. Generally speaking, the present study is well organised. However, some revisions should be conducted, in order to improve the manuscript quality.
In abstract, why authors aimed to investigate the functions of HIF-1α within the ovary of human should be mentioned.
In abstract, the methods section, which tissue samples were used should be mentioned.
Line 67-68, why NDRG1 plays a role in the synthesis of progesterone (P4) in human GCs. It should be explained more.
Section 2.1, which part of ovary was used? The whole ovary? Please explain. How did you maintain the tissues after collection. Please explain.
The antibodies used in the present study, they are monoclonal antibody or polyclonal antibody?
Section 3.1, how many tissues were used for each developmental stage in the present study? Did it have replicate? Please mention it in the section of material and methods.
Please label the cells in Table 2. In addition, why authors did not perform the qPCR analysis of HIF-1α and NDRG1 in various stages of human ovary?
The description in results section should be provided in more details. It is too simple in this format.
In discussion section, line 229-232, the conclusion should be verified by qPCR analysis.
Line 236-279, did any other publications that analyse the functions of NDRG1 in ovarian development in any other species? Please provided and compare.
Author Response
Reviewer #3
Thank you for reviewing our manuscript. We appreciate your constructive comments, which were very helpful in revising our manuscript. We have revised the manuscript following the concerns highlighted by you.
The present study aimed to analyse the functions of HIF-1α and NDRG1 in various stages of human ovary. Generally speaking, the present study is well organised. However, some revisions should be conducted, in order to improve the manuscript quality.
- In abstract, why authors aimed to investigate the functions of HIF-1α within the ovary of human should be mentioned.
Response:
As per your suggestion, we have added an explanation in the abstract regarding the reason for investigating HIF-1α expression in human ovarian follicles.
Page 1 lines 12–14
“However, the timing of HIF-1α expression and its specific biological function in the follicular development of the human ovary remain unclear.”
- In abstract, the methods section, which tissue samples were used should be mentioned.
Response:
Thank you for your pertinent comment. We have described the kind of samples used in this study in the abstract and methods sections.
Page 1 lines 16–18
“We used ovarian tissues from eight women with regular menstrual cycles who were not undergoing hormonal treatment.”
Page 2 lines 90–93
“We evaluated the developmental stages of follicles (i.e. primary, secondary, and tertiary follicles, and corpus luteum) in the ovarian tissues because they contain follicles at various developmental stages.”
- Line 67-68, why NDRG1 plays a role in the synthesis of progesterone (P4) in human GCs. It should be explained more.
Response:
Thank you for your pertinent comment. We have added the following text to explain the role of NDRG1 in the synthesis of P4.
Page 2 lines 67–70
“Notably, our previous study has shown that reducing NDRG1 levels leads to decreased mRNA expression of steroidogenic acute regulator protein (StAR) and CYP11A1 (p450scc), as well as a reduction in P4 production, which suggests that NDRG1 is involved in progesterone synthesis in human GCs [28].”
- Section 2.1, which part of ovary was used? The whole ovary? Please explain. How did you maintain the tissues after collection. Please explain.
Response:
We added the comment regarding the ovarian samples used in this study.
Page 2 lines 80–81
“Formalin-fixed and paraffin-embedded whole ovarian tissues were used for immunohistochemical analysis.”
- The antibodies used in the present study, they are monoclonal antibody or polyclonal antibody?
Response:
Thank you for your pertinent comment. We have added the clonality of the antibodies used in this study.
Page 4 lines 148–152
“rabbit polyclonal NDRG1 antibody (1:1,000; cat. no. #5196: Cell Signaling Technology), rabbit monoclonal StAR antibody (1:1,000; cat. no. #8449: Cell Signaling Technology), rabbit monoclonal CYP11A1 (p450scc) antibody (1:1,000; cat. no. #14217: Cell Signaling Technology) or mouse monoclonal β-actin antibody (1:5,000; cat. no. A5316: Sigma-Aldrich, St. Louis, MO, USA)”
- Section 3.1, how many tissues were used for each developmental stage in the present study? Did it have replicate? Please mention it in the section of material and methods.
Response:
The human ovaries include various developmental stages of the follicles. We used eight human ovarian samples for immunohistochemistry for HIF-1α and NDRG1 to evaluate their expression patterns at the various follicular developmental stages. Accordingly, we have added the following explanation in the text.
Page 2 lines 86–88
“We evaluated the developmental stages of follicles (i.e. primary, secondary, and tertiary follicles, and corpus luteum) in the ovarian tissues because they contain follicles at various developmental stages.”
- Please label the cells in Table 2. In addition, why authors did not perform the qPCR analysis of HIF-1α and NDRG1 in various stages of human ovary?
Response:
Thank you for pointing this out. We have labeled the cells in Figure 2.
We performed the immunohistochemical analysis for HIF-1α and NDRG1 at various human ovarian follicular developmental stages because we hypothesized that HIF-1α and NDRG1 play important roles in development of follicles at each stage. The results of the present study clearly demonstrate that the expression patterns of HIF-1α and NDRG1 vary throughout the development stages of follicles, indicating that HIF-1α activation is the preceding event of NDRG1 expression, and that HIF-1α is crucial for the regulation of NDRG1 expression in the ovary, with both HIF-1α and NDRG1 contributing to the luteinization process.
However, qPCR analysis at various stages of follicular development was not performed in this study, because it is very difficult to extract RNA from human follicles at each developmental stage.
- The description in results section should be provided in more details. It is too simple in this format.
Response:
As per your suggestion, we have revised the results section in detail.
Page 4 lines 172–173
“Table 2 shows the positive rates of HIF-1α protein at various ovarian follicular development stages. ”
Page 5 lines 175–177
“HIF-1α protein was detected in the cytoplasm of more than 70% GCs in primary (Figure 1A) and secondary follicles (Figure 1B), whereas it was barely detectable in the nucleus of GCs.”
Page 6 lines 191–194 and 198
“Table 3 shows the positive rates of the NDRG1 protein at various follicular developmental stages. NDRG1 expression was not observed in the nucleus at all stages of follicular development and restricted to the cytoplasm of GCs. NDRG1 protein was also detected in the cytoplasm of theca interstitial cells."
“Thus, NDRG1 protein expression increased with follicular development.”
Page 7 lines 231–232
“Addition of FSK significantly increased the expression of StAR, CYP11A1 (p450scc), and progesterone.”
- In discussion section, line 229-232, the conclusion should be verified by qPCR analysis.
Response:
qPCR analysis at various stages of follicular development was not performed in this study, because it is difficult to extract RNA from human follicles at each developmental stage.
- Line 236-279, did any other publications that analyse the functions of NDRG1 in ovarian development in any other species? Please provided and compare.
Response:
To our knowledge, no study has examined the functions of NDRG1 in follicular development in any other species. Accordingly, we have included the following statement in the Discussion.
Page 9 lines 288–290
“However, to our knowledge, the expression patterns of NDRG1 and the relationship between NDRG1 and HIF-1α proteins expressions throughout ovarian stages have not been extensively studied in any species, including humans.”
Round 2
Reviewer 1 Report
Comments and Suggestions for Authors
None
Reviewer 3 Report
Comments and Suggestions for Authors
I have no further comment.